# Toward Routing River Water in Land Surface Models with Recurrent Neural Networks

Mauricio Lima[1,2], Katherine Deck[2], Oliver R. A. Dunbar[2], and Tapio Schneider[2]

[1]Ecole Polytechnique
[2]Division of Geological and Planetary Sciences, California Institute of Technology

**Correspondence:** Mauricio Lima (mauriciodemouralima@gmail.com)

**Abstract.** Machine learning is playing an increasing role in hydrology, supplementing or replacing physics-based models. One notable example is the use of recurrent neural networks (RNNs) for forecasting streamflow given observed precipitation and geographic characteristics. Training of such a model over the continental United States (CONUS) has demonstrated that a single set of model parameters can be used across independent catchments, and that RNNs can outperform physics-based models. In this work, we take a next step and study the performance of RNNs for river routing in land surface models (LSMs). Instead of observed precipitation, the LSM-RNN uses instantaneous runoff calculated from physics-based models as an input. We train the model with data from river basins spanning the globe and test it using historical streamflow measurements. The model demonstrates skill at generalization across basins (predicting streamflow in catchments not used in training) and across time (predicting streamflow during years not used in training). We compare the predictions from the LSM-RNN to an existing physics-based model calibrated with a similar dataset and find that the LSM-RNN outperforms the physics-based model: a gain in median NSE from 0.56 to 0.64 (time-split experiment) and from 0.30 to 0.34 (basin-split experiment). Our results show that RNNs are effective for global streamflow prediction from runoff inputs and motivate the development of complete routing models that can capture nested sub-basis connections.

## 1  Introduction

The surface water cycle is a key component of the climate system (Oki and Kanae, 2006), and river routing of runoff from the land to the ocean is an important transport process simulated in land surface models (LSMs) within climate models (Li et al., 2015). Routing models provide a freshwater source to ocean models and have a range of additional applications, covering water resource management (He et al., 2017) to flood hazard assessments under climate change scenarios (Wobus et al., 2017).

River basins, the areas drained by individual rivers and their tributaries, tile the land surface into weakly connected domains, with water transport between basins given by the river streamflow. Two main processes are typically considered in river modeling: hillslope routing and river channel routing (Mizukami et al., 2016). Hillslope routing describes how water moves across the landscape, considering factors such as topography, soil characteristics, and vegetation. It is the process that leads to the time lag between instantaneously generated runoff on land and the aggregation of runoff at a river channel, contributing to streamflow. This process is unresolved in the river models used in LSMs and must be parameterized. In contrast, river channel

routing describes how water moves from upstream channels to downstream outlets within the river network itself. Both the hillslope and river channel routing processes occur simultaneously within a basin.

Within an LSM, a river routing model must demonstrate generalizability across multiple temporal and spatial scales. In time, it should capture the seasonal cycle and the faster surface runoff response to precipitation events. In space, it must display generalizability across basins, one of the main challenges in modern hydrology (Sivapalan et al., 2003; Hrachowitz et al., 2013). What we call regional basin generalizability, known in the hydrology community as regionalization, describes the skill of a model at using the learned characteristics of gauged basins (where streamflow is measured by streamgauges) to predict behavior in other, possibly ungauged, basins, generally in the same region at a sub-continental scale. Several approaches exist to tackle this problem (Prieto et al., 2019). What we call global basin generalizability refers to the same concept, but across all regions, and is also referred to as global-scale regionalization (Beck et al., 2016). In both cases, hydrological models that rely heavily on single-basin calibrations have greater difficulty generalizing due to overfitting to individual basins (Kratzert et al., 2024). Moreover, the vast majority of basins in many regions are ungauged; the generalization of models calibrated with basins from well-represented regions to those with a lack of gauges therefore is especially challenging (Feng et al., 2021).

A physics-based approach to river modeling can be implemented considering different processes at different scales. Lohmann et al. (1996) was one of the first river routing schemes for LSMs, using a linear model to account for hillslope routing in each grid and the linearized Saint-Venant equation to model water transport in between grids. Later models, like the storage-based schemes of Oki and Sud (1998) and Branstetter (2001), used simplified 1D kinematic wave routing (KWR) equations to route water for both hillslope and river channel routing jointly, but explicitly accounting for the spatial distribution of streamflow under various approximations. More advanced approaches (e.g., Ye et al., 2012; Wu et al., 2014), apply 1D KWR explicitly accounting for the differences between hillslope and river channel routing. Moreover, Li et al. (2013) additionally accounted for tributary channels explicitly. More recent models, such as Mizukami et al. (2016), offer flexibility by coupling different possibilities for hillslope and river channel routing in a domain that can be either vector- or grid-based. Physics-based representation of these processes present several advantages. First, the physical equations describing the flow naturally conserve water mass, which is crucial for systems simulated for long periods of time, as is the case in LSMs. Second, the interpretability of the model is straightforward since one knows exactly what physical laws are being used. Finally, physical laws have a long history of success in physical modeling; thus, these approaches are commonly employed in streamflow forecasting models across disciplines. Despite these advantages, however, physical models tend to perform poorly at regional and global basin generalizability, and it has been argued that this is due to challenges in expressing routing processes across scales and locations using simple physical laws (Nearing et al., 2021). For a more detailed overview of physics-based routing models, see Shaad (2018).

Recently, a class of recurrent deep learning models, referred to as Long-Short-Term-Memory (LSTM) models, have outperformed physics-based models on the rainfall-runoff problem (Kratzert et al., 2019b). In Kratzert et al. (2018), an LSTM was trained to model the entire land hydrology system for the CONUS. The model took observed precipitation as input (along with other dynamic inputs such as near-surface temperature, surface pressure, etc.) and then simulated streamflow at the outlet of gauged catchments, implicitly modeling snowpack, soil storage, runoff, hillslope routing, and river channel routing. In Kratzert

et al. (2019b), attributes such as topography, vegetation, and soil properties from different catchments were added to the model to improve its performance. The model did not explicitly account for routing between basins and treated each catchment independently. Nonetheless, the model showed good performance in regional calibrations, where a single set of parameters is learned using data from multiple catchments at once. Further studies also showed that these types of models were successful at basin generalization, predicting streamflow on the outlet of catchments not included in the training set (Kratzert et al., 2019a); more recently, similar models have demonstrated greater global basin generalizability than physics-based models (Nearing et al., 2024). In particular, these results suggest that models can represent the unresolved processes in hillslope and channel routing accurately. A drawback to using these machine learning (ML) models in LSMs is that they are not guaranteed to conserve mass a priori (or produce otherwise physical output, such as positive streamflow); however, constraints can be enforced by adapting the architecture of the network, for example, to be mass conserving (Hoedt et al., 2021). More importantly, though machine-learning based models have been used for routing between basins in specific regions (Moshe et al., 2020), these types of models have not been used yet for this purpose over entire continents, which is a necessary step in order to implement them in LSMs.

## 1.1 Our contributions

Our goal in this work is to explore the use of LSTMs for river routing in a global LSM. To do so, we make multiple alterations to the rainfall-runoff LSTM of Kratzert et al. (2019b), including in model architecture and training and validation procedures:

(i) We use instantaneous runoff (both surface and sub-surface) as input, rather than precipitation, assuming that the runoff is provided by a separate land model. This is done to be able to keep track of where water is stored within the land surface. Keeping track of water fluxes (runoff, evaporation, transpiration) and storage (snow, soil, etc.) is crucial in LSMs for understanding interactions between key land surface components, such as soil, vegetation and snowpack, all of which interact with the atmosphere through energy and water fluxes (Bonan, 2019).

(ii) We construct a globally consistent dataset to train and validate runoff-driven models. We incorporate runoff variables from reanalysis and use a globally unified system of basin characterizations, to ensure that our routing model can be integrated into an LSM in the future. This also requires consistency between gauged catchments and geographical definitions of the nested sub-basins provided by our base dataset (Lehner et al., 2008). A similar idea was first introduced by the Caravan dataset (Kratzert et al., 2023), using precipitation instead of runoff and catchments instead of globally consistent sub-basins.

(iii) We evaluate the trained LSTM models on LSM-relevant tasks, including generalization across time and basins, using both CONUS and global training data. We demonstrate good performance of the model and interpret results at a granular level by breaking down skill over different geographical regions.

(iv) We compare our generalization experiments with the physics-based LISFLOOD model (Van Der Knijff et al., 2010), which underlies the Global Flood Awareness System (GloFAS), an operational product and service of the Copernicus

Emergency Management Service. To do so, we use the GloFAS discharge product provided as reanalysis. Results show the RNN approach displays superior performance.

(v) We consider the mass conservation properties of the LSTM (Appendix D).

This work represents a first step toward using LSTMs for river routing in LSMs; however, we still treat each basin as independent, as in Kratzert et al. (2019b). The problem of routing between basins is left for future work.

## 1.2 Outline and Notation

The terms catchment and basin are often used interchangeably in the hydrology literature. In this paper, we use "basin" (and sub-basin) for the units of topographical subdivision of the world into drainage areas, and "catchment" for the drainage area of
100 specific gauges. Additionally, we use the term generalization in time as described above (generalizing to time periods unseen in calibration, but only for basins used in calibration), in place of what the hydrology community often refers to as performance in gauged basins. We use the term basin generalization as described above (generalizing to basins not used in calibration), in place of what the hydrology community refers to as regionalization or performance in ungauged basins. We choose this terminology because whether a basin is gauged or not is a property of the basin, while time and basin generalizability are
105 experimental design concepts. For example, only gauged basins can be used for validating a model, regardless of whether that model is calibrated in a "gauged" (generalization in time) or "ungauged" (basin generalization) fashion.

The paper is structured as follows. Section 2 presents the various components of our model, including data engineering and the training of the ML model. Section 3 presents our findings regarding time and basin generalization, a comparison to a physics-based model, and an investigation of model performance by basin attributes. Section 4 summarizes the main results
and outlines future directions, including applying the LSTM to routing between basins.

## 2 Methods

### 2.1 Dataset

The first phase of this project consisted of constructing a consistent dataset that allows world-wide calibrations and simulations. Using global data for training is important as it increases the likelihood of generalization outside the training sample. In an
115 LSM, river routing must be simulated across the entire globe, and not just across basins in the training set. To construct the dataset, we need both forcing data, which vary in time and space, as well as static attributes describing physical characteristics of each basin, which are assumed to only vary in space and which encode how runoff is routed within a basin. We additionally require streamflow data in each of these basins, which is the quantity our model is predicting. As explained in the introduction, we only target within-basin routing (hillslope routing and within-basin channels routing), and not main channel routing between
120 nested sub-basins. This allows us to treat each basin as independent in training and simplifies the training task, at the expense of not making use of the information present in the river network structure. In practical terms, this implies that each catchment represented by a gauge in our dataset must have a clear match to a basin.

### 2.1.1 Basins and static attributes

The HydroSHEDS dataset (Lehner et al., 2008) provides a vector-based division of the globe into basins (HydroBASINS, Lehner and Grill, 2013) viewable in levels (1 to 12) of resolution: as the levels grow, basins are subdivided into nested sub-basins, following the topography of the region. Moreover, each basin has static attributes derived from well-established global digital maps (HydroATLAS, Linke et al., 2019), which we use to construct the training data, as explained in Section 2.2. These static attributes are divided into seven sub-classes: Hydrology, Physiography, Climate, Land Cover & Use, Soils & Geology and Anthropogenic Influences. This offers a detailed description of basins that will be used to span the dimensional space in which our model operates. Static attributes are assumed to be constant over time and were chosen based both on previous studies (Kratzert et al., 2019b) and on our physical understanding of the problem. A table with all selected static attributes used in our models is shown in Appendix A. A particularly important static attribute is the area of the catchment, described in 2.1.3.

### 2.1.2 Dynamic inputs

Dynamic variables are the ones that change with time: sub-surface and surface runoff (mass attributes), temperature at 2-m height, surface pressure, and solar radiation over each basin. While in principle only mass attributes are required, additional variables were found to improve the accuracy of the model overall. Dynamic variables such as evaporation from rivers and re-infiltration are ignored. Water that is lost from river channels via these mechanisms is not explicitly tracked but can be implicitly present.

We derive a daily timeseries for each dynamic input for each basin using the grid-based reanalysis dataset provided by ERA5-Land (Muñoz-Sabater et al., 2021). Each point of the grid was attributed to the corresponding basin polygon in space using a simple ray-casting algorithm (Shimrat, 1962). To account for grid cells overlapping with basin boundaries, a Monte Carlo simulation was used to estimate how much of the cell area overlapped with the basin. We added noise to each grid point coordinate and computed the probability that a point is found inside the polygon. Figure 1a illustrates the process.

All dynamic variables are calculated daily. Sub-surface and surface runoff are extensive variables and are summed over the set of grid points inside each basin. The air temperature at 2-m height, the surface pressure, and the solar radiation are intensive variables and are averaged over the set of grid points inside each basin. Spatial averaging is applied to all variables within a basin, involving the division of the cumulative timeseries values within each basin by the corresponding number of grid points within that basin. We highlight that this process is feasible starting from any grid resolution, which makes the model adaptable to other datasets without the need for recalibration.

A final pre-processing step prior to training and network evaluation is to normalize all input variables, following Kratzert et al. (2022).

### 2.1.3 Streamflow

Measurements of streamflow as a function of time were obtained from the Global Runoff Data Centre (GRDC, https://portal.grdc.bafg.de/, last accessed 11 September 2024). To associate the discharge records from the river gauges, identified by latitude and longitude,

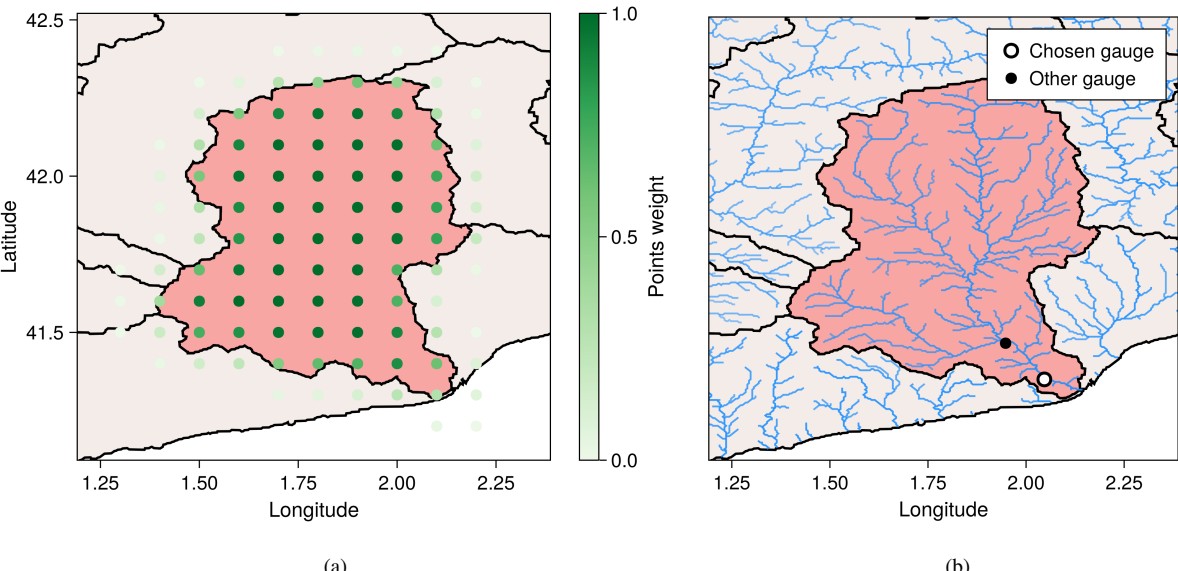

**Figure 1.** (a) Illustration of the Monte-Carlo algorithm used for transforming gridded ERA5-Land data into basin-specific data for HydroSHEDS basin 2070017000 (shaded pink, located at the east coast of Spain). Grid cells that are completely inside the basin have a weight of 1 because their entire area lies within the basin. Grid cells outside the basin have weights less than 1, representing the fraction of their area within the basin. Other basins are shaded grey, and their boundaries are outlined in black. The white area is the sea (in this case, *Mar des Baleares*). (b) GRDC gauges and the river network structure within the same basin. In the figure, the chosen gauge (white circle) is the one used in the calibration of our model for this specific basin, as it better represents the entire drainage area of the basin. The other gauge has a smaller catchment area, so it represents a smaller fraction of the behavior of the basin.

with their corresponding basins, we employed the ray-casting algorithm to determine which polygon (basin vector) encloses each gauge. We found that in many cases, the gauge catchment area defined by GRDC and the basin area for the corresponding basin in HydroSHEDS were not in agreement. This can occur for two main reasons:

- When a single gauge catchment area contains several small sub-basins, the catchment is much larger than the basin containing the gauge, which is probably a sub-basin dependent on upstream basins inside the catchment.

– When a basin has smaller, secondary rivers with associated gauge catchment areas within it, the basin is much bigger than the gauge catchment within it, which probably corresponds to a sub-basin.

     To reduce errors arising from assigning gauge catchment areas to the incorrect basin, we used a filter allowing not more than a 20% difference between the catchment and basin area. This value was chosen as a balance between a small threshold (favoring gauges closer to the outlet of the basin, hence more representative of the outflow of the basin) and a large threshold

(favoring more matching examples, hence more data for calibration). Ultimately, this filter has the role of choosing only basins with good spatial agreement with their gauge catchment. Moreover, only gauges with more than 1 year of consecutive data

were considered in the training phase. This procedure was inspired by Sutanudjaja et al. (2018). We highlight that not all basins have data in the test phase, which means that some basins can be used for training but not for testing. The result of this process is shown in Figure 1b.

The streamflow measurements provided by GRDC are in terms of the local time zone where the gauge is located. To match with ERA5-Land reanalysis, we interpolated these timeseries to UTC. As the GRDC timeseries have daily increments, this shift assumes a constant streamflow during the day, which is a coarse approximation. Furthermore, we take the gauge catchment area defined by GRDC (and not the basin area defined by HydroSHEDS) to use as static attribute for the gauge linked to a basin for model training. This variable is crucial for the model because it enables it to adjust the input runoff, normalized to

the basin area, according to the size of the drainage area, in order to predict streamflow. All other static and dynamic inputs are estimated using the corresponding basin defined in HydroSHEDS.

The resulting runoff and streamflow timeseries on the basin in Figure 1 is shown in Figure 2.

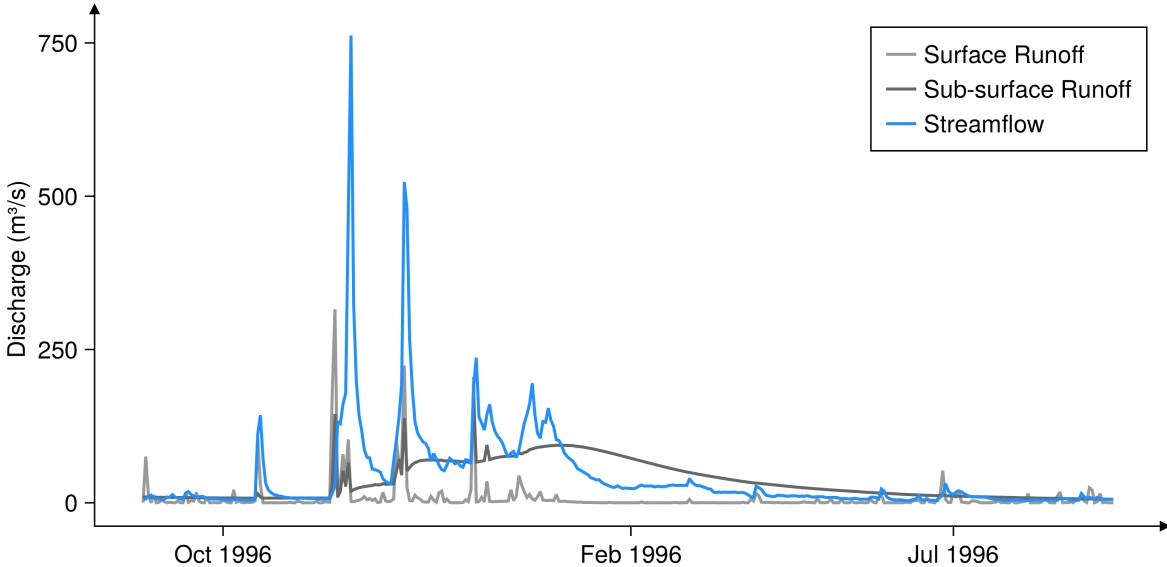

**Figure 2.** Surface, sub-surface runoff, and streamflow during one year for basin 2070017000 of HydroSHEDS, located at the east coast of Spain.

### 2.1.4    Dataset summary

The process was applied for all 9 continental areas and for 3 of the 12 levels available in HydroSHEDS (levels 5, 6, and

7). These levels were chosen based on their sub-basins area to be relevant for typical resolutions of climate models. Table 1 summarizes the number of catchments in each level both in the CONUS and globally, and reports their areas. The study covered

the time span from 1990 to 2019, with certain gauges exhibiting gaps in discharge data. No gap filling technique was applied, and only original values were retained.

**Table 1.** Relevant statistics for gauged basins in the US and over the Globe, including the number of catchments and their median, minimum, and maximum catchment areas.

| Level | US | | | Global | | |
|---|---|---|---|---|---|---|
| | Number | Median (km$^2$) | Min, Max (km$^2$) | Number | Median (km$^2$) | Min, Max (km$^2$) |
| Level 05 | 55 | 2.9e4 | 4.9e3, 2.4e5 | 224 | 3.0e4 | 3.6e3, 3.4e5 |
| Level 06 | 150 | 1.1e4 | 1.9e3, 6.2e4 | 443 | 9.9e3 | 1.2e3, 7.0e4 |
| Level 07 | 193 | 4.4e3 | 7.6e2, 2.1e4 | 660 | 3.7e3 | 3.9e2, 5.4e4 |

## 2.2 LSTM model

Recurrent neural networks are a subset of neural networks that contain recurrent connections and were designed to handle sequential data (Rumelhart et al., 1986; Goodfellow et al., 2016). RNNs are particularly effective when the predictor requires a (possibly long) time history data to make accurate forecasts. This is relevant to our use case, as the key variables (surface and subsurface runoff) are provided as a daily timeseries for each basin, but streamflow may depend on the time history of runoff because of physical storage and transport processes. Given an element $x_t$ in a sequence $(x_1, ..., x_T)$ indexed by discrete time

$(1, ..., T)$ and a set of learnable parameters $\theta$, a hidden state $h_t$ in a typical RNN is completely described by the recurrent relation

$$h_t = F(h_{t-1}, x_t; \theta), \tag{1}$$

where $F$ represent some flexible parametrized model, and this equation is subject to an initial condition $h_0$. Here, $x_t$ is the vector concatenating the vector of dynamic variables ($x_t^d$) and the vector of static attributes ($x^s$): $x_t = (x_t^d, x^s)$. At the final

time $T$, the corresponding output $\hat{q}_T$ (daily streamflow in $\mathrm{m^3\,s^{-1}}$) depends on the hidden state through

$$\hat{q}_T = G(h_T; \theta), \tag{2}$$

where $G$ is a function that is composed of a linear transformation and a dropout layer used to prevent overfitting (Srivastava et al., 2014). A schematic of the network is shown in Figure 3.

LSTMs (Long Short-Term Memory; Hochreiter and Schmidhuber, 1997) are a type of RNN with a specific general structure

of the function $F$. A detailed explanation of the LSTM network used in this work can be found in Appendix B. The design of the network avoids the vanishing gradient problem that plagues the training procedures for vanilla RNNs, so that optimization of the weights of an LSTM is far more effective. As an RNN, it also allows for the state at previous steps to affect the output at the current step.

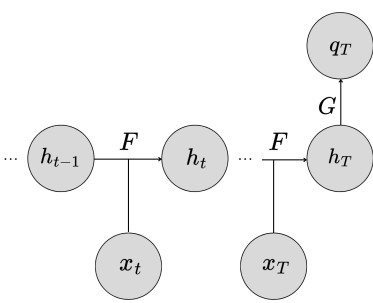

**Figure 3.** Diagram of dependencies in the recurrent neural network. The input vector $x_t = (R_t^s, R_t^{ss}, ...; A, ...)$ is a concatenation of dynamic inputs, such as instantaneous surface runoff $R^s(t)$, sub-surface runoff $R^{ss}(t)$, and other dynamic inputs which vary with time $t$, and static attributes such as the catchment area $A$, and others. The variable $h$ denotes the hidden state. (Figure modeled after and inspired by those in Goodfellow et al. (2016).)

The training procedure seeks to optimize the loss function by varying the free parameters of the LSTM. Our loss function $\mathcal{L}$
is a modified Nash-Sutcliffe efficiency (NSE), defined as

$$\mathcal{L}(\boldsymbol{\theta}) = \frac{1}{B} \sum_{b,t} \frac{(\hat{q}_{b,t} - q_{b,t})^2}{(s(b) + \delta)^2}, \tag{3}$$

where $\hat{q}_{b,t}$ and $q_{b,t}$ are the predicted and observed streamflow at basin $b$'s outlet for time $t$ in the training set, $s(b)$ is the standard deviation of streamflow at basin $b$ and $\delta$ is a small number (set to $0.1$) for numeric stability. The loss function is averaged by the number of basins in the training set $B$. Moreover, we chose to skip missing values of streamflow to increase the number
of valid samples upon which we can compute the loss, which may then vary for each basin. Since there is no factor in the numerator dividing by the number of observations per basin, this indicates that basins with more observations (fewer gaps) are effectively weighted more by the loss function. We refer the reader to Kratzert et al. (2019b) for further discussion of this loss function choice. Note that the NSE is a standard metric for performance used in timeseries prediction; we provide a further description and analysis of the NSE metric in Section 2.3.
To tune the model, we used the Adam optimizer (Kingma and Ba, 2014), with the recommended parameters of $\beta_1 = 0.9$, $\beta_2 = 0.999$ and $\epsilon = 10^{-8}$. Following (Kratzert et al., 2019b), we chose the number of recurrent iterations of the LSTM to be $T = 270$, the number of features in the hidden state $h$ to be $256$ and the dropout probability of the linear layer in $G$ to be $0.4$. We trained the model for 35 epochs, with a learning rate of $10^{-3}$ for the first 10 epochs, $10^{-4}$ for the 10 following epochs, and $10^{-5}$ for the last 5 epochs. More details can also be found in our code (Lima, 2024).
Without additional constraints, there is no guarantee that the LSTM model conserves water mass (aside from indirectly, by matching observed streamflow given runoff as input). An important consideration for LSMs is how to adapt this model in order to conserve mass, possibly accounting for processes like evaporation from rivers and re-infiltration of water into the soil.

Possible approaches include changing the loss function or adapting the neural network design to be mass conserving (Hoedt et al., 2021); we leave such adaptations for future work.

## 2.3 Metrics

To assess the performance of the model, we use the Nash-Sutcliffe efficiency (NSE, Nash and Sutcliffe, 1970), which, for a given outlet, can be written as

$$\text{NSE} = 1 - \frac{\sum_t (\hat{q}_t - q_t)^2}{\sum_t (q_t - \bar{q})^2}, \tag{4}$$

where $\hat{q}_t$ and $q_t$ are the predicted and observed streamflow at a basin outlet for time $t$ and $\bar{q}$ is the averaged observed streamflow over all valid times in the validation or test set. Note that here we omit the index $b$ because the expression is computed for each basin, and not for an ensemble of basins as we did in the expression of the loss function. In this expression, 1 is a perfect score ($\hat{q}_t = q_t$), and it gets worse (lower NSE) as the fraction of the mean squared error ($\sum_t (\hat{q}_t - q_t)^2 / n$) normalized by the variance of the streamflow ($\sum_t (q_t - \bar{q})^2 / n$) increases, where $n$ indicates the number of observations for each basin. This normalization implies the NSE to lie in $(-\infty, 1]$. Note that if a model is predicting only the mean flow at the outlet (i.e., $\hat{q}_t = \bar{q}$), we would have an NSE of 0. We will use this value as reference to evaluate performances above a naive mean flow baseline (NSE > 0) vs. performances below a naive mean flow baseline (NSE < 0), as suggested in Knoben et al. (2019).

Another commonly used metric in hydrology is the Kling–Gupta efficiency (KGE, Gupta et al., 2009), which, for a given basin, can be written as

$$\text{KGE} = 1 - \sqrt{(r-1)^2 + (\alpha - 1)^2 + (\beta - 1)^2}. \tag{5}$$

In the equation, $r$ is the linear correlation coefficient between simulated and observed timeseries; $\alpha = \hat{\sigma}/\sigma$ is the variability ratio, given by the ratio between the standard deviation in simulations $\hat{\sigma}$ and the standard deviation in observations $\sigma$; $\beta = \hat{\mu}/\mu$ is the bias ratio, given by the ratio between the mean in simulations $\hat{\mu}$ and the mean in observations $\mu$. This score represents an explicit decomposition of the normalized mean squared error into the three components $r$, $\alpha$, and $\beta$. The score likewise lies in $(-\infty, 1]$, with 1 being a perfect score. We can define a reference value based on the KGE for a model predicting only the mean flow. In this case, we would have no correlation ($r = 0$), no variability ratio ($\alpha = 0$), but a perfect bias ratio ($\beta = 1$), which gives a reference KGE of $1 - \sqrt{2} \approx -0.41$. We will use this reference value as a parameter to evaluate performances above a naive mean flow baseline (KGE > $1 - \sqrt{2}$) vs. performances below a naive mean flow baseline (KGE < $1 - \sqrt{2}$).

As both scores are unbounded along the negative axis, outliers can lead to large negative values. Therefore, we use the median score (rather than the mean) over the entire ensemble of basins to quantify the results. In discussion of only well-performing basins (outperforming the meanflow reference), it is still useful to use the mean; we denote this metric as "Mean$_{\text{NSE}>0}$" for the NSE and "Mean$_{\text{KGE}>1-\sqrt{2}}$" for the KGE. Better models will have this mean closer to 1. We also define the fraction of poor-performing basins (worse than the meanflow reference) in the test set. We denote this metric as "%$_{\text{NSE}<0}$" for the NSE and "%$_{\text{KGE}<1-\sqrt{2}}$" for the KGE. Better models will have this fraction closer to zero. As both metrics are normalized, we can compare river discharges from basins with different sizes and regimes in different climates.

## 3  Results

We present a series of experiments carried out to quantify the performance of the model. We compare the behavior of a model driven with precipitation to that of one driven with runoff, and we compare the behavior of a model trained and tested in the USA with one trained and tested globally. To assess generalizability across basins and times, we experiment with different training/testing/validation splits. The model's training time varied with the datasets and the longest run took a few hours with one V100 GPU. Furthermore, we compare the performance of the model against the physics-based model LISFLOOD (Van Der Knijff et al., 2010), provided by GloFAS v4.0 reanalysis data, which is publicly available in the Copernicus Climate Data Store (https://confluence.ecmwf.int/display/CEMS/GloFAS+v4.0, last accessed 4 September 2024). The simulations generated by both models are presented under various conditions, including those where both models produce poor scores, those where the simulations are close to the median score, and those where each model demonstrates good performance relative to the median score of each model. An additional point is made regarding possible comparisons with the models presented here and other LSTM models in the literature. Finally, we analyze the performance by continent and other attributes. An analysis by HydroSHEDS levels is provided in Appendix C. All models shown in this section are based on the same LSTM architecture, which is not strictly mass conserving. An analysis of mass conservation for our LSTM can be found in Appendix D.

### 3.1  From precipitation in the USA to runoff worldwide

This series of experiments investigates the performance of LSTMs trained and validated using basins in the USA only and LSTMs trained and validated on global data. The USA was chosen as reference for being a well characterized region in terms of data availability and because it allows for a more direct comparison to previous results (Kratzert et al. (2019a, b); see Section 3.2.2 for further discussion). Our goals were to determine how the performance of the model changes when the dynamic input changes (from precipitation to runoff driven from reanalysis), to determine how the model performance changes when trained with more varied data (from the USA to the global case), and to investigate how the model performs at the basin and time generalization tasks.

### 3.1.1  Time generalization

To address these questions, we begin with a time training/validation split. The dataset was divided into three different timeseries for all basins: from 01/10/1999 to 30/09/2009 for training, from 01/10/2009 to 30/09/2019 for validation, and from 01/10/1989 to 30/09/1999 for testing. We compared the model trained on the USA with runoff input with the model trained on the USA with precipitation input. We also compared the performance of the model trained on the USA with runoff input with the model trained using the global dataset with runoff input.

Figure 4 shows the results of the models trained in this time-split configuration (solid lines). Shown is the cumulative density function (CDF) of the NSE scores, truncated to $[0, 1]$, for the 5 different models analyzed. Since a perfect model has an NSE score of 1, the best models have a CDF that remains nears zero at all values of the NSE except at 1. The median NSE corresponds to $\mathrm{CDF} = 0.5$.

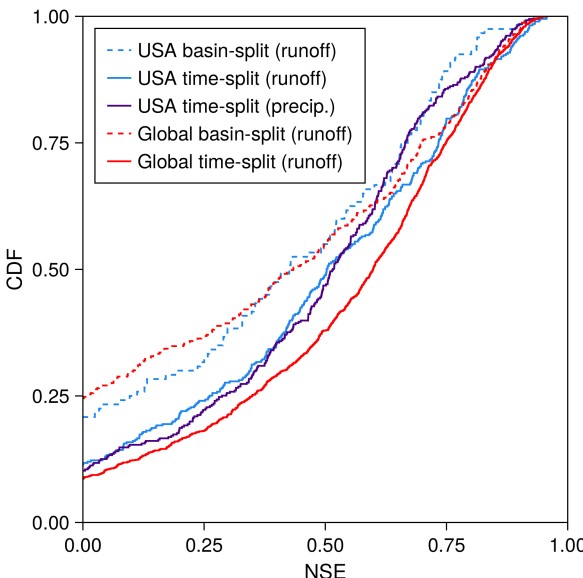

**Figure 4.** Cumulative NSE density function for the LSTM models trained on different datasets. The models in blue and red were trained using runoff as input; the model in purple was trained using precipitation as input. Solid lines indicate time-split datasets; dashed lines indicate basin-split datasets. The experiments are described in detail in Section 3.1.1 and Section 3.1.2.

The results of the time-split experiment show that the model exhibits similar performance when we change the dynamic input data from precipitation to runoff. Quantitatively, we observe a median NSE of 0.50 for the runoff-driven model and a median NSE of 0.52 for the precipitation-driven model when both are trained on USA data. The model exhibits an increase in accuracy when trained with global data, yielding a median NSE of 0.60. This suggests that the model exhibits enhanced learning capabilities when trained on a more diverse dataset.

### 3.1.2  Basin generalization

We next investigate LSTMs trained on random subsets of all basins globally and test on a disjoint set of all basins globally. We divide our dataset of basins by choosing 70% for training (from 01/10/1999 to 30/09/2009) and validation (from 01/10/1989 to 30/09/1999) and 30% for testing (in the same time window as the training set). More precisely, from a total of 398 (USA) and 1327 (global) basins matched with gauges after the filters were applied, we use 278 (USA) and 928 (global) to train and validate and 120 (USA) and 399 (global) to test the model in this configuration. In this second set of experiments, we only compare models driven by runoff.

The results of the basin-split experiments are also shown in Figure 4 (dotted lines). The results demonstrate that the basin-split models perform more poorly compared with their counterparts trained with a time-split. This is expected as the unseen basins problem is a more challenging task. However, similar performances are observed in both the regionally calibrated (USA)

and globally calibrated models: a median NSE of 0.43 is observed for both cases. In Appendix C, it is shown that the global basin-split model has a better performance for higher levels (05 and 06) in comparison to level 07, which means that the basin generalizability of the model does depend of the general size of the basins.

### 3.1.3 Summary of results

Table 2 highlights some important additional points. The time-split configuration trained on observed USA precipitation data has the smallest fraction of performances with scores worse than the naive mean-flow baseline model among the configurations trained and tested in the USA, suggesting that it suffers less from outliers and that there is room for improvement in runoff modeling. We also highlight that $Mean_{NSE>0}$ is a good quantitative number to summarize the overall behavior of the curves in the CDF in Figure 4.

**Table 2.** Performance metrics for LSTM models over different datasets.

| Model | NSE | | |
|---|---|---|---|
| | $\%_{NSE<0}$ | $Mean_{NSE>0}$ | Median |
| USA basin-split (runoff) | 20.83% | 0.51 | 0.43 |
| USA time-split (runoff) | 11.25% | 0.54 | 0.5 |
| USA time-split (precip.) | 10.23% | 0.53 | 0.52 |
| Globe basin-split (runoff) | 24.31% | 0.54 | 0.43 |
| Globe time-split (runoff) | 8.62% | 0.58 | 0.6 |

## 3.2 Comparison with other models

### 3.2.1 Comparison with a physics-based model

In a second series of experiments, we compare the LSTM model with the LISFLOOD model, provided by GloFAS v4.0. This choice was made because LISFLOOD explicitly calculates runoff and performs routing of river water from reanalysis data from ERA5 (Hersbach et al., 2020). Moreover, the comparison between the LSTM and LISFLOOD was straightforward to carry out without additional calibrations, as the LISFLOOD streamflow predictions are provided with the reanalysis data. That said, there are some important differences between our model and training procedure and those of GloFAS: (i) GloFAS uses different forcings from ERA5, including precipitation, to calculate runoff internally, whereas the LSTM uses runoff variables computed by the HTESSEL model (Balsamo et al., 2009) in ERA5-Land (ii) LISFLOOD is a complete river routing model, whereas the LSTM models basins independently; (iii) the objective function for GloFAS is based on the modified KGE metric whereas the LSTM was trained to optimize the NSE; and (iv) GloFAS is calibrated with a different global set of gauges.

Below, we compare the results of GloFAS to the LSTM in both a time- and basin-split experiment. However, several caveats should be noted. First, a sensible time-split experiment is challenging, as GloFAS draws different dates for training, validating

and testing for each basin. This means that the validation time frame of the LISFLOOD model may coincide with the test period
used in our comparison here, potentially simplifying the prediction task of the time-split experiment for GloFAS relative to
the LSTM. Second, the locations of only GRDC gauges that were used to calibrate GloFAS are known and other datasets may
have been used in place of the same gauges (GloFAS team members, private communication). This allows for a basin-split
experiment comparison, but (a) some of the gauges in our test set may have been used in the calibration of GloFAS, and (b)
the unknown time-split may mean that for some of the test basins, GloFAS is generalizing in both space and time. The former
makes the GloFAS prediction task easier than the LSTM task, while the latter makes the GloFAS prediction task harder than
the LSTM task, since the LSTM is trained and tested in different basins but using the same time window. For a large enough
temporal training period and for a global set of basins, we would expect that generalizing in space and time is similar to
generalizing in space, since that is the much more challenging task. For all of these reasons, we note the basin- and time-split
experiments with a "*" in the Table and Figure presented in this Section.

In order to construct the set of gauges used in our comparisons, we considered the following:

- When carrying out the time-split experiment, we restrict the analysis to the intersection of the test set of basins used by
  our global time-split model and the basins used to calibrate GloFAS.

- When carrying out the basin-split experiment, we restrict the analysis to the intersection of the test set of basins used by
  our global basin-split model and the basins not used to calibrate GloFAS.

- We filter basins by allowing no more than a 20% difference between the catchment area reported by GRDC and the
  catchment reported by GloFAS.

The process resulted in a total of 283 basins in the time-split and 197 basins in the basin-split configurations. We highlight
that these basins are independent of other basins, since our models do not route water between basins. The LSTMs are evaluated
against GloFAS on the same time periods.

Figure 5 shows the comparison between the LSTM trained in the time-split and basin-split configurations and the physics-
based benchmark. Figure 5a shows the CDF for the NSE score, which is the score that our model was designed to optimize.
Figure 5b shows the CDF for the KGE score, which is the score optimized by GloFAS.

For the gauges we use, the LSTM shows an overall better accuracy compared with GloFAS in both basin- and time- split
experiments. For the temporal split, the LSTM has a median score of 0.64 on NSE and 0.72 on KGE, whereas GloFAS displays
a median score of 0.53 on NSE and 0.71 on KGE. For the basin split, the LSTM has a median score of 0.33 on NSE and 0.41
on KGE, whereas GloFAS displays a median score of 0.30 on NSE and 0.40 on KGE. Observe that despite being optimized for
NSE, the LSTM has similar scores on both basin- and time-split to GloFAS on KGE. Moreover, the LSTM predictions result
in fewer basins with performance worse than the baseline in all experiments. We show other metrics in Table 3.

It is interesting that the LSTM time-split model performs better here compared with that of Section 3.1. This may suggest
that the set of basins used by GloFAS contains gauges with more reliable measurements.

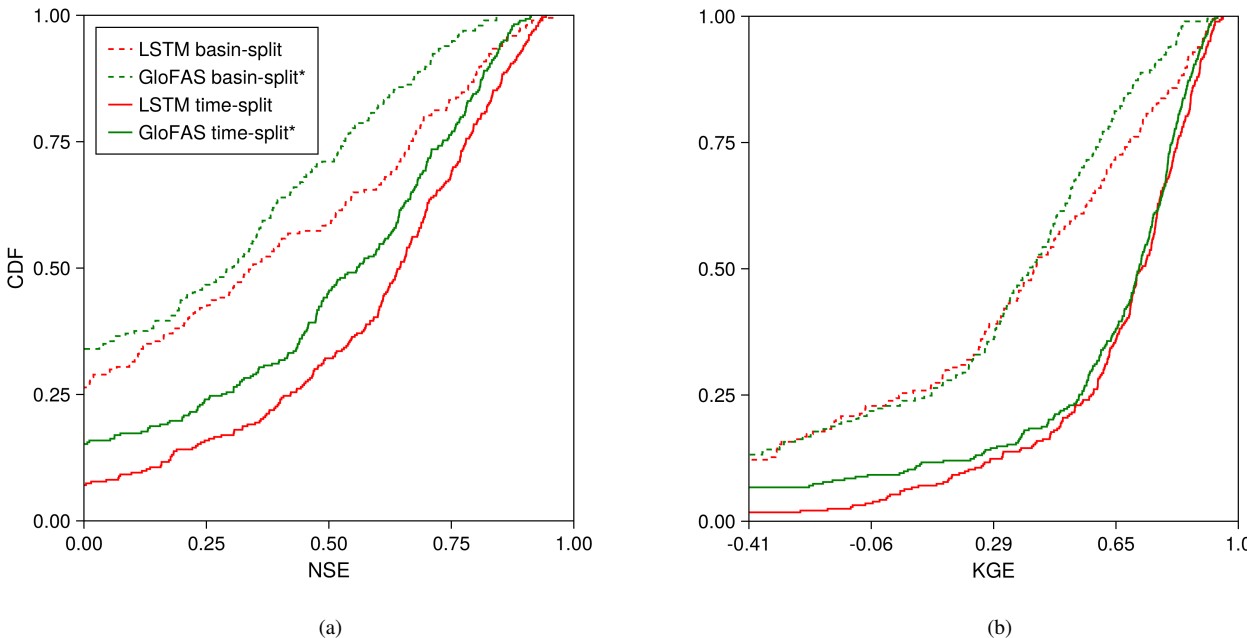

(a)                  (b)

**Figure 5.** Cumulative density functions for (a) NSE and (b) KGE for the LSTM and GloFAS. The domain is truncated to $[0, 1]$ for NSE and to $[1 - \sqrt{2}, 1]$ for KGE, with the lower bounds corresponding to the mean-flow prediction reference value. The legend in (a) equally applies in (b). The "*" is there to make it explicit that GloFAS experiments are not exactly a basin-split nor a time-split, as we do not know the exact set of dates used to calibrate the model. As detailed in Section 3.2.1, this can potentially simplify the tasks performed by LISFLOOD.

**Table 3.** Performance metrics for the LSTM model in basin-split and time-split configurations and for the benchmark reanalysis from GloFAS.

| Model | NSE | | | KGE | | |
|---|---|---|---|---|---|---|
| | $\%_{\mathrm{NSE}<0}$ | $\mathrm{Mean}_{\mathrm{NSE}>0}$ | Median | $\%_{\mathrm{KGE}<1-\sqrt{2}}$ | $\mathrm{Mean}_{\mathrm{KGE}>1-\sqrt{2}}$ | Median |
| LSTM basin-split | 26.4% | 0.51 | 0.34 | 12.18% | 0.44 | 0.41 |
| GloFAS basin-split* | 34.01% | 0.45 | 0.30 | 13.2% | 0.41 | 0.40 |
| LSTM time-split | 7.07% | 0.62 | 0.64 | 1.77% | 0.66 | 0.72 |
| GloFAS time-split* | 15.19% | 0.58 | 0.56 | 6.71% | 0.66 | 0.71 |

### 3.2.2 Comparison with other LSTM models

In this section we discuss possible comparisons to similar LSTM-based models from Kratzert et al. (2019b) and Nearing et al. (2024). These comparisons are in general difficult to make as each model is trained using different basins, using different

dynamic input variables, and, in the case of Nearing et al. (2024), with different experimental designs. The models are also intended for different use cases.

In Kratzert et al. (2019b), an LSTM model was trained using observed dynamic inputs gathered for catchments in the CONUS (Addor et al., 2017). In a time-split experiment, Kratzert et al. (2019b) reported a median NSE of 0.73 (without ensembles). In our time-split experiment for the USA with reanalysis precipitation as input (Section 3.1), we found a median NSE of 0.52. Importantly, the dynamic inputs of the former come from observations, while the dynamic inputs used to train the latter come from reanalysis data. As a consequence, these two models are not easily comparable as using observed precipitation (among other inputs) is more accurate than reanalysis data. Additionally, our LSM-RNN was calibrated with fewer basins.

In Nearing et al. (2024), an LSTM model was trained using a rich set of dynamic inputs coming from multiple datasets including high-resolution forecasts, estimates from satellites, and ERA5-Land reanalysis data. Precipitation, and not runoff, was used. The purpose of the model is for forecasting floods in watersheds globally. The model was training globally, and was tested in a time-and-basin split simultaneously. Due to large differences in input data and experimental design, a comparison between the published results of this model and our model is not necessarily meaningful to make. Using more input data with higher accuracy than reanalysis data should improve streamflow predictions, and hence we expect this model to outperform our LSM-RNN.

## 3.3  Simulations vs. Observed streamflow

In this section, we present simulated timeseries along with observations for various values of NSE and KGE to get a visual sense of performance. For four different basins, Figure 6 shows the predictions of the LSTM model in the global time-split configuration, the GloFAS reanalysis, and the observed discharge at the corresponding GRDC gauge.

In the first example (Figure 6a), we have a performance that lies around the median NSE performance of our model: (0.61/0.63) for the LSTM and (0.52/0.56) for GloFAS in the time interval displayed. The next example (Figure 6b) shows a poor performance in both NSE/KGE scores for both the LSTM (0.02/−0.2) and for GloFAS (−0.02/−0.61). For both models, this is due to underprediction of a peak in the streamflow. It is of note that this basin produces zeros discharge for much of the time interval, and the peak discharge measures only $24\mathrm{m}^3/\mathrm{s}$, several orders of magnitude smaller than the other basins in the figure—we come back to trends in the model performance with basin characteristics in the following section. The third plot (Figure 6c) shows an example where the LSTM has a very good performance (0.76/0.68) and outperforms GloFAS (0.37/0.08). We observe that, although GloFAS displays good correlation, the LSTM is quantitatively closer to observations. The last plot (Figure 6d) shows a case where GloFAS (0.84/0.70) outperforms the LSTM model (0.79/0.73) in the NSE score. One can infer that the LSTM underestimates the bigger peak, but can overestimate the lower peaks, while GloFAS shows opposite behavior.

## 3.4  Geographic patterns

Lastly, we investigate the performance of the LSTM model trained with global runoff data in the time-split configuration, for different basin characteristics and geographic properties.

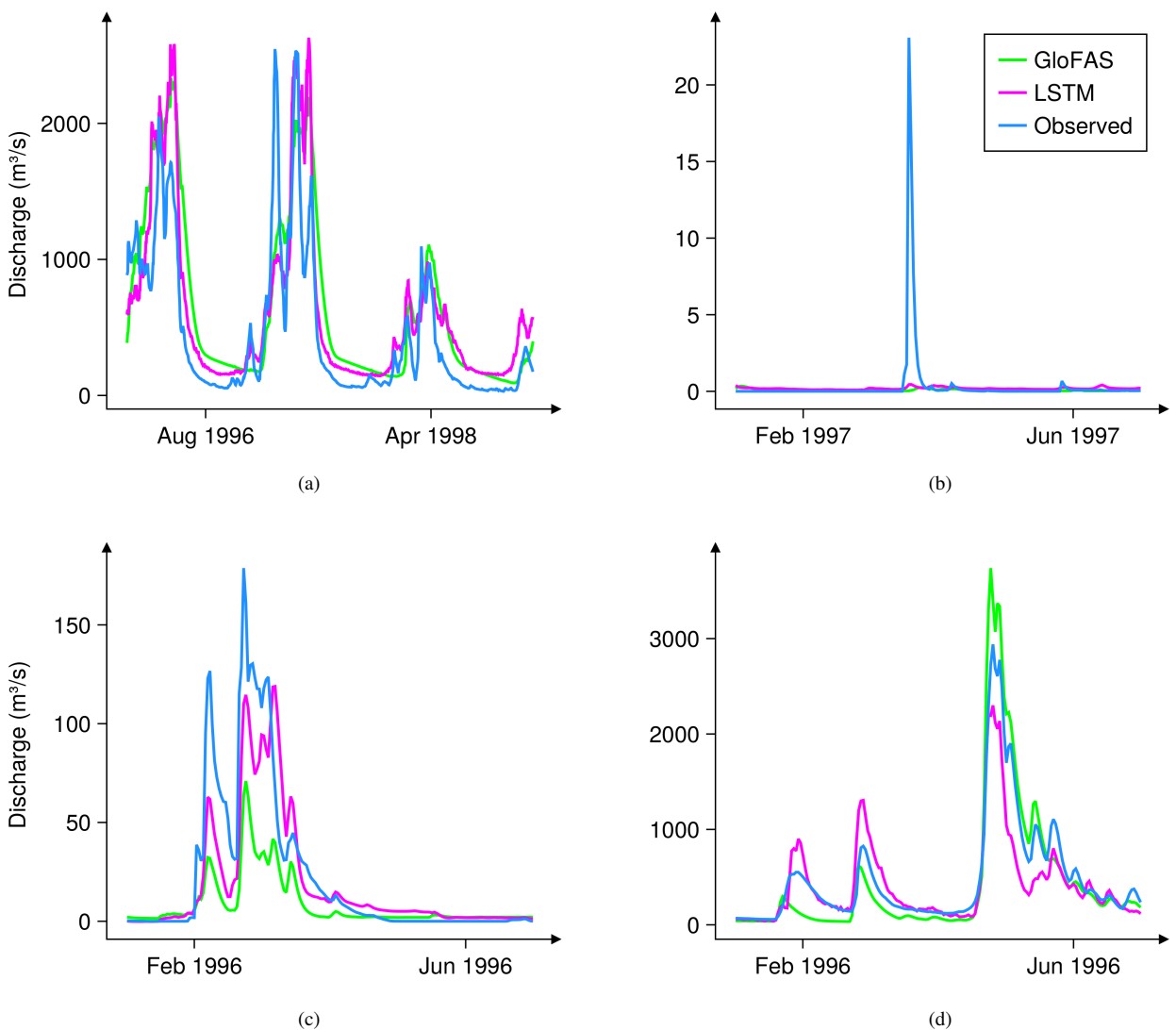

**Figure 6.** Timeseries simulated by the LSTM model (pink), by GloFAS reanalysis (green), and the observed timeseries from GRDC gauges (blue) in four different basins. (a) Basin 6050344660 in HydroSHEDS level 05, located in Brazil, with gauge located at the Itacaiúnas River. (b) Basin 1061638580 in HydroSHEDS level 06, located in South Africa, with gauge located at the Seekoei River. (c) Basin 7050013170 in HydroSHEDS level 05, located in California, with gauge located at the Salinas River. (d) Basin 7060363050 in HydroSHEDS level 06, located at the border between Canada and the United States, with gauge located at the Saint John River.

Table 4 shows statistics of NSE scores for each continental basin. We can see that regions that are under-represented in the data (such as Africa and Australasia) have worse scores, but that under-representation in terms of location is not enough to predict poor performance. In particular, the basins in Asia are well modeled from only 38 examples, and Siberia is well-modeled from only 2 examples. These results may be explained by the similarity of climate conditions between these regions and other well-represented regions. For example, conditions may be similar between Siberia and the Canadian Arctic.

**Table 4.** Performance metrics for the LSTM model, trained with global runoff data in the time-split configuration, over the 9 continental basins defined by HydroSHEDS.

| Continental basin | NSE | | | Number of basins |
|---|---|---|---|---|
| | $\%_{NSE<0}$ | $Mean_{NSE>0}$ | Median | |
| Africa | 25.33% | 0.3 | 0.19 | 75 |
| Europe and Middle East | 4.32% | 0.67 | 0.71 | 301 |
| Siberia | 0.0% | 0.72 | 0.72 | 2 |
| Asia | 5.26% | 0.56 | 0.59 | 38 |
| Australasia | 9.41% | 0.49 | 0.47 | 85 |
| South America | 10.0% | 0.57 | 0.58 | 170 |
| North and Central America | 10.04% | 0.56 | 0.54 | 528 |
| Arctic (northern Canada) | 0.0% | 0.69 | 0.7 | 101 |
| Greenland | - | - | - | 0 |

The global distribution of NSE scores for the gauges in Table 4 can be seen in Figure 7. From the map, we see the contrast between data-rich regions (North America and Europe) and data-poorer regions (East Asia and central and northern Africa).

We also investigate the aridity index, which is provided in the HydroATLAS dataset (Linke et al., 2019) following the dataset of Zomer et al. (2008). The aridity index is given by the ratio between the mean annual precipitation and the mean annual evapotranspiration, and is calculated on a per grid cell basis. Hence, lower values of aridity index represent arid climates and higher values represent humid climates, for further discussion, one may reference to the newest version of the dataset in Zomer et al. (2022). In our analysis, the aridity index has been found to be a strong predictor of the model's performance. Figure 8 shows a trend toward poorer performance in drier regions (i.e., regions with lower aridity index), consistent with findings in Feng et al. (2020), but now on a global scale. (For context, the basin depicted in Figure 6b has an aridity index of 0.26.) This trend extends to other variables, such as mean runoff, for which the model also shows poorer performance in drier regions.

## 4 Discussion and conclusion

Long short-term memory models have been shown to be the state of the art for modeling streamflow in hydrological systems (Kratzert et al., 2019b), but most studies using these models have so far been restricted to specific regions, and they have

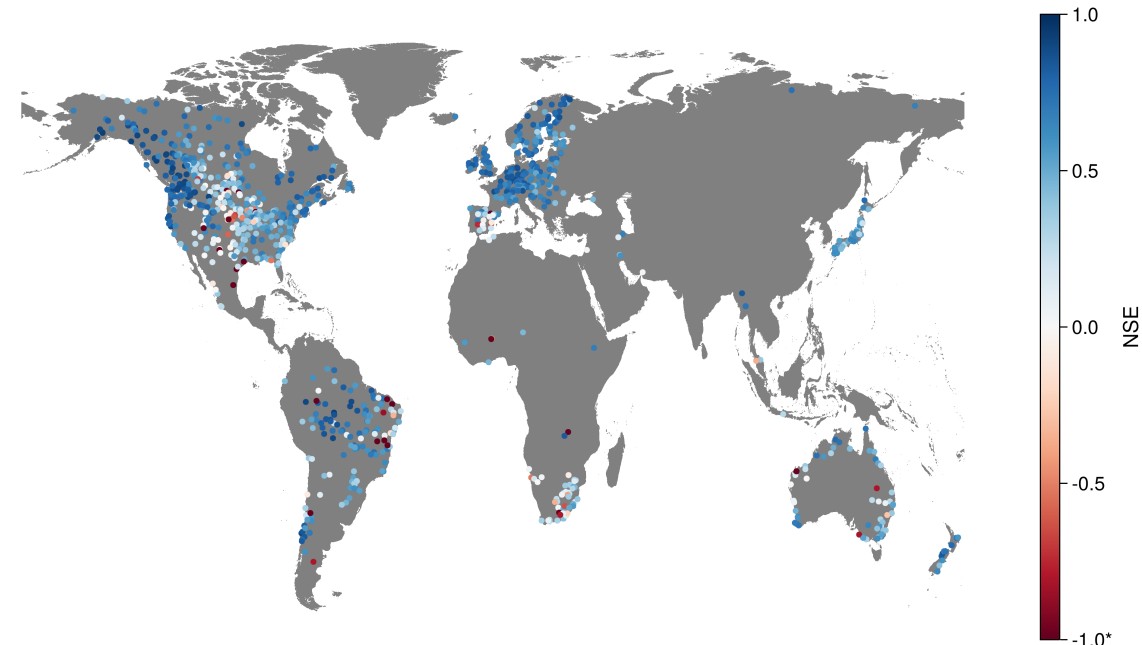

**Figure 7.** Distribution of NSE scores for the LSTM model, trained globally in the time-split configuration. Each point corresponds to the location of the gauge linked to the basins in HydroSHEDS levels 05, 06, or 07. The "*" is used to clarify that the range of NSE values is clipped to greater than $-1.0$. We assigned the same color to all basins with a score less than $-1.0$ for simplicity.

focused on representing the entire land hydrological system as a single entity (Kratzert et al., 2019b; Koch and Schneider, 2022). However, for integration into the land surface models used in weather forecasting and climate modeling, it is crucial for river models to demonstrate proficiency in generalizing across basins and time scales globally, using surface and subsurface runoff data, rather than precipitation data. This necessity arises because traditional river models in land surface models route water, but leave the modeling of snow, soil, and vegetation hydrological processes to other model components. In this study, we have taken concrete steps toward developing an ML model that can substitute traditional river routing models within LSMs.

We have trained and validated an LSTM at the task of predicting streamflow from modeled runoff worldwide. We began our analysis by contrasting the results from a precipitation-driven model, which holistically represents land hydrology, with those of a runoff-driven model. For the United States, we found that an LSTM trained only to route runoff performs comparably to one designed to simulate the entire land hydrology system, even when trained on potentially less accurate and biased runoff data. Our investigation of the model's generalization capabilities revealed that a globally trained model, as opposed to one trained exclusively using data from the USA, achieved superior performance when both were fed runoff data in a time-split configuration. This improvement underscores the potential benefits of incorporating diverse global data. However, when evaluating models trained in the basin-split configuration, the performance gain was not as pronounced, highlighting the complex challenge of global basin generalization. This challenge is exacerbated by imbalances in global data availability and

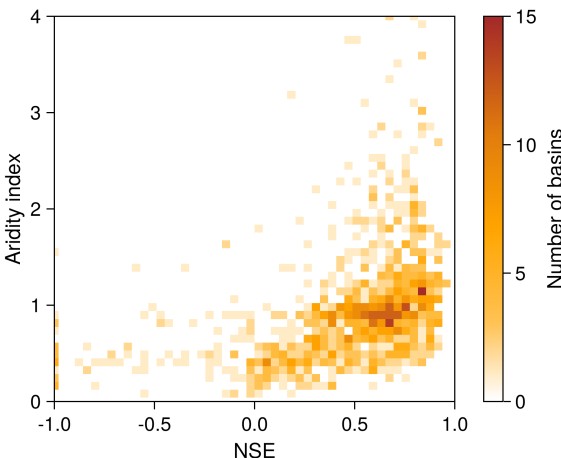

**Figure 8.** Aridity index (a static attribute provided by HydroATLAS) versus NSE scores for the LSTM time-split model. Each small square in the figure represent the amount of basins within the corresponding ranges of NSE and aridity index in the test set. The aridity index is defined as the ratio between mean annual precipitation and mean annual evapotranspiration, hence lower values represent drier basins. We have observed that lower NSE scores preferentially are found in more arid basins.

the wide range of possible climate conditions across the globe, which may not be well sampled in geographically localized training data (e.g., from the USA).

Additionally, our analysis revealed a correlation between the model's performance and the aridity index. While streamflow in arid basins can be modeled well by the LSTM, it is also true that all basins with a poor NSE have a lower aridity index. This suggests that drier regions pose challenges for the LSTM model, but that other basin features may affect performance as well. The dependence on aridity may be due to the model's difficulty in capturing short-term precipitation events that trigger streamflow peaks in these areas(Feng et al., 2020). Moreover, the ERA5-Land reanalysis is driven with precipitation from ERA5, known to have biases in the tropics (Lavers et al., 2022), which could lead to biases in the runoff of ERA5-Land in these regions. As a consequence, our river model may be learning to correct these biases, in addition to routing water. This is a possible pitfall for machine learning models trained with model output and not with observed data. The LSTM model also underperformed in regions likely underrepresented in the training data, such as in the African continent. Despite these challenges, the model's time generalizability was consistent across different basin sizes (HydroSHEDS levels), even if better basin generalizability was observed for bigger basins (Appendix C). We highlight that the model does not need to be re-calibrated for the evaluation of different basin levels, which correspond to different spatial resolutions of LSMs.

We also took steps to compare our model performance with the performance of other models. In general, this is challenging due to differences in experimental design, loss function choice, and data used. We highlight two main conclusions we drew from our comparisons:

1. Using a similar dataset, experimental design, and LSTM architecture, we found that a model trained with reanalysis data (our USA time-split precipitation model) had a significantly worse performance (a drop in median NSE from 0.73 to 0.52) compared to the LSTM of Kratzert et al. (2019b), which was trained with observations. This indicates, unsurprisingly, that modeled variables (e.g., precipitation or runoff from reanalysis) suffer from larger errors than observed precipitation, and that these errors affect the quality of the predicted streamflow. An LSTM trained using modeled runoff, for which observations are not available, may learn both to predict streamflow and to correct biases in modeled runoff. Such biases may differ significantly in different regions of the world, and an LSTM provided with terrain particularities and climate characteristics of each basin from static attributes should be capable to use this information to correct runoff model error. This is undesirable, since it prohibits us from using streamflow data to calibrate our runoff models. However, for our use case of a river model routing water within an LSM, and not modeling the entire land hydrology system, this may be unavoidable.

2. The physics-based LISFLOOD model (GloFAS) is more similar to our model in that it uses modeled runoff (calculated internally) as a dynamic input to perform streamflow routing. We have compared the models on a subset of the gauges used to calibrate GloFAS (time-split) and on a subset of gauges not used for calibration (basin-split). The LSTM model showed more accurate simulations in both scenarios: a gain in median NSE from 0.56 to 0.64 (time-split experiment) and from 0.30 to 0.34 (basin-split experiment). Despite these averages differences, GloFAS shows superior fidelity to observations than the LSTM in some cases, while the LSTM is superior in others. However, there are still some differences in experimental design between the two. A recalibration of GloFAS using our data and experimental design for a direct one-on-one comparison is beyond the scope of this work.

In order to integrate our model with an LSM, it must be extended to enable inter-basin channel routing, and it must respect physical principles such as mass conservation. Even if approaches showing how to account for inter-basin connections in a restricted region with ML models have appeared (Moshe et al., 2020), physical models such as LISFLOOD for routing between basins are currently the standard for LSMs. One may interpret our work as the hillslope routing component of a complete routing model. In that case, two possible ways to extend this to route water between basins are to use a physics-based channel routing model, or to use similar neural networks to additionally perform river channel routing. For the latter, designing training strategies to mitigate error accumulation across connected basins may be necessary. Moreover, incorporating architectural adjustments to ensure mass conservation (e.g., Hoedt et al., 2021), is a necessary step for incorporation in climate models.

In conclusion, we have demonstrated that a ML-based river model exhibits basin and time generalizability, a requirement for use in a global climate model. Our study presents the first step toward using such a model in the LSM component of climate models as well as for short-, medium-, and long-range hydrological forecasting using runoff from LSMs within weather forecasting models. The results presented here motivate further research to extend these models for comprehensive river routing.

*Code availability.* All data used to generate our training and test data are open source. The source code for the data engineering, the models, the extraction from the benchmark, and the visualizations can be found in https://github.com/limamau/Rivers.

# Appendix A: List of static attributes used in the models

Table A1 lists the static attributes used in our models. The area comes from GRDC's catchments, all other variables come from the HydroATLAS dataset. All these variables are assumed to be constant in time.

**Table A1.** List of static attributes used in the models

| Variable Name | Description | Units |
| --- | --- | --- |
| pre_mm_syr | mean precipitation | mm |
| ari_ix_sav | aridity index | — |
| area | GRDC catchment's area | $km^2$ |
| ele_mt_sav | mean elevation | m |
| snw_pc_syr | snow percent cover | — |
| slp_dg_sav | mean slope | — |
| kar_pc_sse | karst percent cover | — |
| cly_pc_sav | clay percent cover | — |
| pet_mm_syr | mean potential evaporation | mm |
| for_pc_sse | forest percent cover | — |
| snd_pc_sav | sand percent cover | — |
| slt_pc_sav | silt percent cover | — |
| gwt_cm_sav | ground water table depth | cm |
| run_mm_syr | land surface runoff | m |
| soc_th_sav | organic carbon content | t/ha |
| swc_pc_syr | soil water content | — |
| sgr_dk_sav | stream gradient | dm/km |
| cmi_ix_syr | climate moisture index | — |

# Appendix B: LSTM architecture

As explained in Section 2.2, the LSTM model is a recurrent neural network, where one cell is used recursively for a sequence length $T$ of iterations. In the text, we have represented this cell as a flexible parameterized model $F$. The equations that describe

the internal mechanism of such a function (i.e., a forward pass in an LSTM cell at the $t^{\text{th}}$ iteration) are:

$$\boldsymbol{i}_t = \sigma(\boldsymbol{W}_i \boldsymbol{x}_t + \boldsymbol{U}_i \boldsymbol{h}_{t-1} + \boldsymbol{b}_i), \tag{B1}$$

$$\boldsymbol{f}_t = \sigma(\boldsymbol{W}_f \boldsymbol{x}_t + \boldsymbol{U}_f \boldsymbol{h}_{t-1} + \boldsymbol{b}_f), \tag{B2}$$

$$\boldsymbol{g}_t = \tanh(\boldsymbol{W}_g \boldsymbol{x}_t + \boldsymbol{U}_g \boldsymbol{h}_{t-1} + \boldsymbol{b}_g), \tag{B3}$$

$$\boldsymbol{o}_t = \sigma(\boldsymbol{W}_o \boldsymbol{x}_t + \boldsymbol{U}_o \boldsymbol{h}_{t-1} + \boldsymbol{b}_o), \tag{B4}$$

$$\boldsymbol{c}_t = \boldsymbol{f}_t \odot \boldsymbol{c}_{t-1} + \boldsymbol{i}_t \odot \boldsymbol{g}_t, \tag{B5}$$

$$\boldsymbol{h}_t = \boldsymbol{o}_t \odot \tanh(\boldsymbol{c}_t), \tag{B6}$$

where $\boldsymbol{i}_t$, $\boldsymbol{f}_t$, $\boldsymbol{g}_t$ and $\boldsymbol{o}_t$ are the input, forget, cell and output gates, respectively. Each of these gates have their learnable weights for both the inputs $\boldsymbol{x}_t$ (represented by the matrix $\boldsymbol{W}$) and hidden states $\boldsymbol{h}_t$ (represented by the matrix $\boldsymbol{U}$), as well as learnable biases (represented by the vector $\boldsymbol{b}$). Observe that these learnable parameters don't depend on time – even if the result of each gate may depend on time with $\boldsymbol{x}_t$ and $\boldsymbol{h}_t$, they use the same learnable parameters, which are applied recursively. The input, forget and output gates are enclosed by the sigmoid function $\sigma$ and the cell gate is enclosed by the hyperbolic tangent function $\tanh$. Furthermore, in the expression of cell state $\boldsymbol{c}_t$ and the hidden state $\boldsymbol{h}_t$, $\odot$ represents an element-wise multiplication. Figure B1 schematizes these equations inside the LSTM cell.

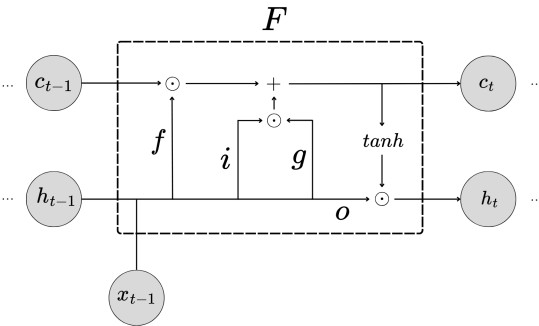

**Figure B1.** Diagram of an LSTM cell, as defined in (B1) – (B6).

Observe that, in comparison to Figure 3, we explicitly represent the cell state $\boldsymbol{c}_t$, which is also a "hidden" state that influence subsequent iterations. The general idea of the LSTM is to add or remove information of this cell state through gates, which is what is written in (B5). On the left side of the equation ($\boldsymbol{f}_t \odot \boldsymbol{c}_{t-1}$), the forget gate $\boldsymbol{f}_t$ (which is enclosed by a sigmoid function) can remove (resp. add) information from the cell state $\boldsymbol{c}_t$ by multiplying each element of the vector by 0 (resp. 1). On the right side of the equation ($\boldsymbol{i}_t \odot \boldsymbol{g}_t$), the input gate applies the same logic to the cell gate to remove or add information to the cell state. For more intuition behind this architecture, we point out to the comprehensive text in https://colah.github.io/posts/2015-08-Understanding-LSTMs/ (last accessed 13 August 2024).

## Appendix C:  NSE by HydroSHEDS level

In this section, we show the the distribution of NSE scores for each of the three HydroSHEDS levels used in the training. The three levels differ in the typical size of basin areas. In Figure C1b, the consistency of the model is re-assuring because it shows that the model is able to adapt to different basin sizes under the same training set. In spite of that, Figure C1a shows that the model generalizes better to unseen basins for larger basins, as it has higher scores for levels 05 and 06 in comparison with level 07 - even if the latter is more represented in the training set. The LSTM in the basin-split configuration slightly outperformed GloFAS under the NSE metric when only evaluated in levels 05 and 06 – a median of 0.47 was observed against a median of 0.45. Even though, GloFAS was still better under the KGE metric – a median of 0.47 against 0.60.

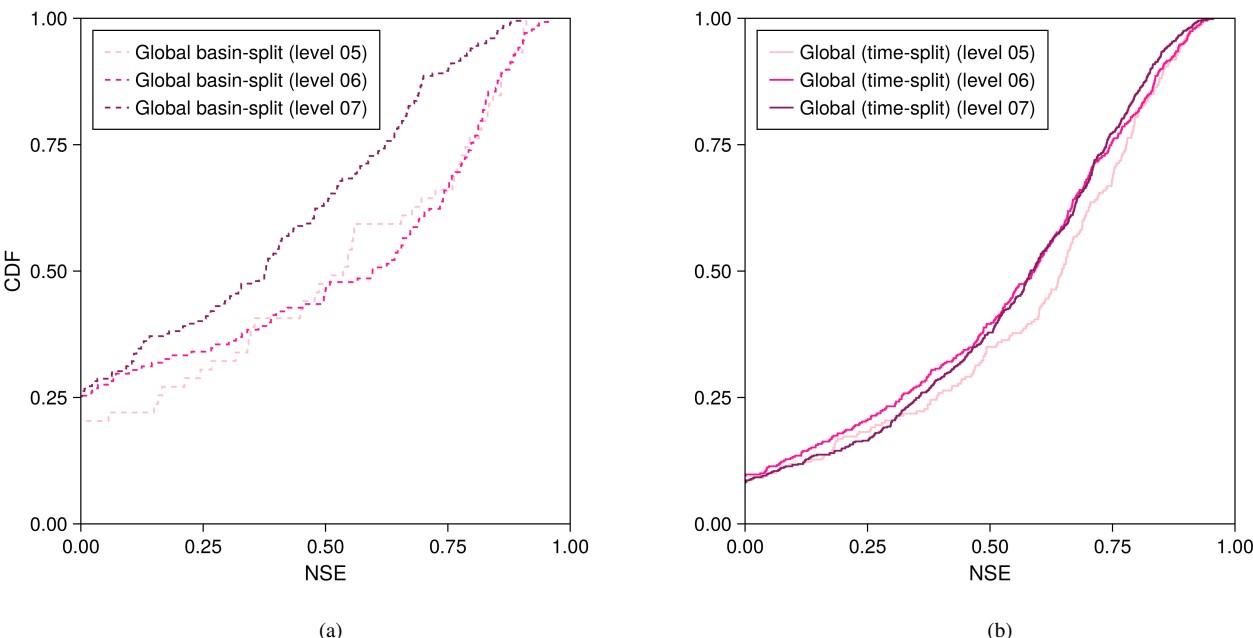

(a)                                                                                    (b)

**Figure C1.** Cumulative density functions for the NSE score of the LSTM in (a) basin-split and (b) time-split configuration for each of the 3 HydroSHEDS levels used in the training set (05, 06 and 07).

## Appendix D:  Mass balance

If there is no loss of water due to infiltration into deeper layers in the ground or evaporation into the atmosphere, the sum of runoff over the area of a catchment should match the discharge at the outlet when averaged over extended periods. We calculate the absolute difference between the outflow from each basin (normalized by its area) and the total runoff within the corresponding catchment over a 9-year time window. The results of this calculation are presented in Figure D1 for streamflow (in blue): simulated by the LSTM (Figure D1a), recorded by observations from GRDC gauges (Figure D1b), and provided by

the GloFAS reanalysis (Figure D1c). In each case, the number of outliers ("N of outliers") represents the quantity of data points falling outside the boundaries of the graphs. We highlight that the runoff calculated internally by LISFLOOD was not used here, but rather the one provided by ERA5-Land, which was calculated by HTESSEL – a different model forced with similar ERA5 data. Under these circumstances, the LSTM conserves mass at a level comparable to the observations and a physical

model. While this is a positive result, it implies that additional processes like evaporation from rivers, re-infiltration of water into the ground, and human interference may need to be understood and modeled in order to achieve a closed water balance. One could expect to find a better mass conservation from the physics-based model, but it should be noted that GloFAS has parameters which control mass loss (to underground storage, for example) and the actual state of mass conservation can vary depending on the version of runoff data that was used in their calculations.

We also depict the relative difference between the definitions of upstream area and catchment area for each of these gauges (in pink). These relative differences are generally small. This implies that there is a good correspondence between the upstream areas used by GloFAS for the calculation of streamflow from runoff inputs and the area used in this study (provided by GRDC).

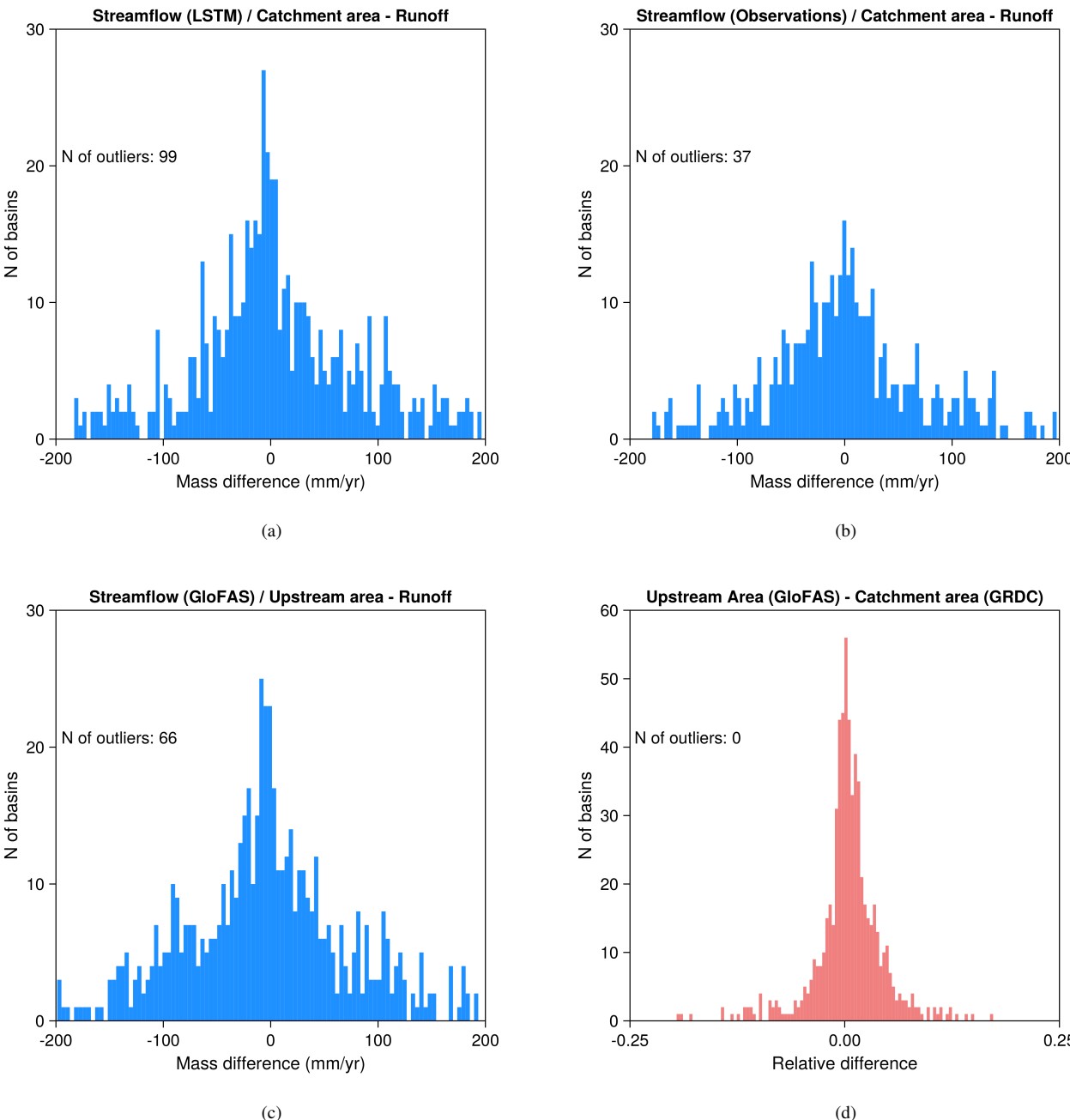

**Figure D1.** Histograms of mass balance in blue and relative difference between area definitions in pink.

*Author contributions.* ML, KD, ORAD and TS: conceptualization, formal analysis, visualization, writing.

*Competing interests.* The authors declare no competing interests.

*Acknowledgements.* This research was supported by the generosity of Eric and Wendy Schmidt by recommendation of the Schmidt Futures program. We thank Frederik Kratzert for a helpful early conversation, as well as the other maintainers of neuralhydrology, for the clear and helpful code. We thank the CESM GloFAS team for providing the set of gauges from GRDC used to calibrate the model that was used to download the discharge reanalysis data, as well as helpful clarifications about GloFAS versions. ML acknowledges Akshay Sridhar, who assisted with technical parts of the code, data handling and insightful conversations, as well as Thomas Dubos for helping this research

project to take place. All numerical calculations and model training used for this manuscript were performed with the help of Caltech's Resnick High Performance Computing Center.

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
