# Peer review of "Toward Routing River Water in Land Surface Models with Recurrent Neural Networks"

_EGUsphere, 2024_

## Author Comment (AC1)

Response to Referee 1. Comments are in black, responses are in blue.

Introduction – The various concepts of the context are explained well and are interesting. However, the review of existing literature related to these concepts is almost completely missing. A good introduction should include key concepts of the problem at hand, i.e. water routing (which is addressed in the current version*). It should then review what has been done so far (both classical and AI-based methods) related to these concepts, thereby revealing the current gap(s) to which your paper would contribute (this is poorly addressed in the current version).
* Although this section could be more engaging by concisely explaining water routing ideas in physics-based models.
We have added further description of physics-based models and highlighted advantages and disadvantages of both classical and AI approaches. We hope that the immediately following "Our Contribution" makes it clearer how our approach addresses certain gaps and what is left to be done in future work.

Methods – Important LSTM training details are missing. For example, the loss function, with its full definition, is a crucial element of the optimization algorithm and should be presented in the Methods section, not in the Metrics section (and it is not sufficient to refer the reader to another work for its definition). The optimization algorithm and the LSTM architecture are also completely missing here and throughout the paper.
In the Methods section (2.2), we added the definition of the loss function and described the optimizer and hyperparameters. We agree that the LSTM architecture should be described. However, because the design of the LSTM architecture was taken from previous work as cited, we chose not to highlight it in the main text, but instead modified Appendix B now to include a detailed description. We reference Appendix B in the main text. Additionally, we modified some notation in Section 2.2 and Figure 3 to make the paper consistent end-to-end.

Benchmark – In its current form, the comparison with LISFLOOD is not fully justified in my opinion, as the existing LISFLOOD simulations were conducted under a different setup that seems to be unknown to the authors. Such simulations involve several subtleties that need to be carefully managed; otherwise, any conclusions drawn would be biased. Why don't the authors conduct these simulations themselves under controlled conditions corresponding to their LSTM experiments?
In our opinion, conducting their simulations is beyond the scope of this paper. For example, it is documented in Alfieri et al. (2020) that a full calibration exercise took 2 months on a HPC Cluster. Moreover, Nearing et al. (2024) conducted similar comparisons as to the ones we have, also without recalibration. That said, we agree that the comparison can be improved, and we have attempted to do so.

In our comparison, we require knowledge of the GloFAS training protocol and predictions at specific gauges. However, we originally only had access to the GloFAS predictions on a grid, and hence we had to associate these grid points with nearby gauges. This left room for uncertainty as to whether we picked the correct gauge. To improve the comparison, we have emailed their team and received the set of gauges from GRDC used for the calibration of the model, which we used to adapt Section 3.2.1.

With this data we were able to separate two experiments in GLOFAS, which we refer to as basin-split* and time-split*. We made clear that both experiments are slightly different from our own basin- and time-split experiments, but they are similar and likely biased towards GloFase, as we document. We think the comparison has a good place in the article as it demonstrates the generalizability skills of the LSTM.

Results (and Discussion) – The analysis of the results does not appear to be sufficiently in-depth, particularly in relation to the few previous regionalization studies using LSTMs over the US continent. There should be a thorough discussion comparing the findings of this study with those from previous research to highlight the contributions and significance of your work, or, to explain any potential divergence from their results (this is missing in the current version).

We have gone through the literature again regarding application of LSTMs to the streamflow prediction problem. We found that LSTMs in previous studies use a different set of gauges and different input datasets (observed precipitation and observed atmospheric conditions, instead of reanalysis data), and this makes the direct comparison between them challenging to interpret. In Section 3.2.2, we discuss this and compare our results with other LSTM models.

The Use of the Term "Forecast" – Based on the content, this paper is not about forecasting but rather about prediction (simulation). This error should be corrected (this mistake does not appear in the Conclusion, where it correctly states: "We have successfully trained and validated an LSTM for the task of predicting streamflow from runoff worldwide").

We have adjusted the text throughout.

MINOR COMMENTS

- PDF Version – The PDF version of the paper did not include line numbers, which made it very impractical for review.

We agree with the referee and apologize for the inconvenience. This happened because we posted our preprint both via EGU and via arXiv. EGU recommends having a single preprint, but arXiv doesn't allow papers with line numbers.

- P.2, Introduction – The phrase "common ungauged basins" in the sentence "This indicates that information in large-scale hydrological datasets is sufficient for generalization tasks, especially to the common ungauged basins (Nearing et al., 2021)" needs clarification. What does "common ungauged basins" mean in this context?

We rephrased this part to make the text more clear.

- Table 1 – Please specify the range for each level presented in the table.

We interpreted this comment as providing the range (minimum and maximum) catchment area for each level. That information is now included in Table 1. Additionally, we note the median catchment size in Table 1, and contrast with the median size of each basin as noted in the text (Section 2.1.4).

- Page 9, Line 6 – Remove "mean" in "mean squared error," as the term does not include any averaging.

We altered the text for clarity. The numerator in the NSE metric per basin is the mean squared error, and the denominator the variance, but the 1/n factor cancels out between the two. Our loss function is slightly different, as explained.

- Appendix B – What are the tested values for each of the three hyperparameters? This information is important and concise enough to be included in the main text.

We have changed Appendix B to provide a detailed explanation of the LSTM architecture. The values of the hyper-parameters were moved to the main text. Note that we have not run extensive tuning experiments with them, but rather used the values used in similar studies, which we have now explicitly referenced.

- Figure 5 – Place the legend above the subplots as it applies to both of them.

We have added "The legend in (a) equally applied in (b)." at the end of the figure's caption.

- Figure 6 – (Maybe) Place the labels (a), (b), (c), (d) inside the respective subplots to save space.

We decided to keep the labels out of the plots as it gives more liberty to the editor.

- P. 16 – It would be interesting if you could present the top 4-5 attributes that, according to your results, show a relationship with model performance. For instance, in which regions are variables like "karst percent cover" and "groundwater table depth" explanatory to some degree in terms of model performance?

We agree that this is an interesting question. In our study, we used many of the same static features that were used in the LSTM model of Kratzert et al (2019b), who did carry out a detailed feature importance study. However, as noted, their model represents the entire land hydrology system, while our model does not. Because of this, there are likely some differences in relative feature importance between the two.

- Figure 8 (caption) – Provide the definition of the aridity index both in the caption and in the text. For instance, you mention  "Drier regions" (i.e., regions with lower aridity index)," but it is natural to expect that the higher the index of a region, the more the climate lacks effective moisture. Additionally, the following sentence in the caption should be stated more carefully: "There is a tendency for worse scores for smaller aridity indexes (i.e., drier basins)," since at the same range of aridity indices, there are basins with good NSE values. Also, state what each point represents in the figure.

We have defined the aridity index and referenced the original and most recent work from the authors of the dataset containing it. We have also rephrased "There is a tendency for worse scores for smaller aridity indexes (i.e., drier basins)," to "We have observed that lower NSE scores preferentially are found in more arid basins.". We also provided further details about the meaning of each small square ("points") in the figure.

- P.18 – "However, it is not clear if this increase in performance is due to a change in the LSTM model." How is Nearing et al.'s LSTM model different from yours? This is an example of studies that should be included in the literature review. The provided context and highlighted differences can then be used as an element of result analysis in your discussion section.

We provide a comparison of our model results to other LSTM models (including Nearing et al., 2024) in Section 3.2.2.

- P.18 – The equivalency between gauged and time-split configuration, as well as between ungauged and basin-split, should be mentioned at their first introduction. Additionally, consider using the terms "gauged" and "ungauged" instead of "time-split" and "basin-split," as these are far more intuitive.

In our understanding, the terms `gauged' and `ungauged' denote a physical feature of a basin, while `time-split` and `basin-split` indicate an experimental design, and there is not a direct correspondence between these concepts. For example, only gauged basins - with measured streamflow - are used across all of our tests, while a model trained in either a time-split or basin-split configuration can be used to predict streamflow in a gauge or ungauged basin.

The terms time-split and basin-split correspond to the notions of temporal and basin generalizability introduced in the introduction. We consider the terms time/basin-split to be technical enough to only include them in the Results section. We have checked the manuscript to make sure that our notation is clear and consistent throughout, and added a small section on Notation in the introduction to make sure the reader is aware of our convention.

- P.18 – The conclusion "suggesting that drier regions pose unique challenges for the LSTM model" is incorrect. As mentioned above, many of your basins with good NSEs fall within the same aridity interval. Please revise this conclusion to reflect the actual results.

We rephrased this conclusion as follows: "Additionally, our analysis revealed a correlation between the model's performance and the aridity index. While streamflow in arid basins can be modeled well by the LSTM, it is also true that all basins with a poor NSE have a lower aridity index. This suggests that drier regions pose challenges for the LSTM model, but that other basin features may affect performance as well."

- P.19 – Remove the parentheses around "Hoedt et al., 2021."

We fixed this.

- I find the mass balance analysis interesting. You may consider placing it inside the main body of the paper.

We kept the mass balance analysis in the appendix as we felt it was more secondary, but we appreciate the referee's comment.

---

## Author Comment (AC2)

Response to Referee 2. Comments are in black, responses are in blue.

**Main points**

**Pg3, bullet iv: Which version of GloFAS is used in the paper?** Version 2 and previous were forced by runoff from the ECMWF land surface model ECLand (formally, HTESSEL) with river discharge produced by the LISFLOOD river routing scheme, whereas from version 3, the full LISFLOOD hydrological model was used. This is quite important given the premise of the benchmarking here. If using version 2 (forced with HTESSEL + LISFLOOD river routing) then this work benchmarks only the river routing part, but if GloFAS version 3 or newer, then GloFAS is driven by a full LISFLOOD hydrological model (forced with e.g. Precipitation and temperature from ERA5, rather than runoff). See the GloFAS version system changes and associated documentation for details: https://confluence.ecmwf.int/display/CEMS/GloFAS+versioning+system
**Again on Pg 10, line 5-6:** If GloFAS v3 onwards, then it uses the fully physical-based LISFLOOD model, if using GloFAS v2 it uses runoff from ECMWF ECLand + the LISFLOOD river runoff scheme.

In our work we used GloFAS version 4 and we have now made that explicit. While it's true that in this version the full LISFLOOD is used to generate runoff and route river water, we think that it is still a good benchmarking because the runoff is calculated explicitly from a physics-based model, as is HTESSEL. We have tailored the text to make this distinction as clear as possible.

**Pg 4, 2nd line from bottom: GloFAS is forced with ERA5 not ERA5-Land,** while there is not strong differences, ERA5-Land does show better performance for hydrological modelling (see Munoz-Sabater et al. (2021) for a hydrological benchmark on GloFAS with both). These differences impact the benchmarking in Lima et al. (2024) and the details and differences in experimental set-up must be qualified.
We have made this distinction clear in point "(i)" of Section 3.2.1.

**Pg 8, last para, line 5-6:** This seems to be where the detail of the ML model as used in this paper is outlined. It's constrained to the Appendix B with details found in code uploaded to Zenodo. While I strongly support and compliment the authors for uploading their code, I still think there is insufficient detail and considerations of limitations explained within the main manuscript. The method is very difficult to repeat without more detail explained to the reader. Figure 3 outlines the RNN, but where are the assumptions/limitations for this particular river routing application, where are the detail on the model training method/time periods/temporal resolution used here etc.

We agree with the referee and we have changed appendix B to include a detailed explanation of the LSTM architecture. Section 2.2 now describes all the hyperparameters used to both setup the RNN architecture and to train it. The temporal resolution of the dynamical inputs are stated in this same section. We hope that with these modifications it's now easier to repeat the method.

**Pg 10; line 7:** what do you mean by "hindcasts"; hindcast is used in the forecast literature to mean running forecasts for past dates. However, I do not believe you are running forecasts here.
We took the term out of the text and adjusted where needed.

**Pg 16; line 6:** In the performance of models for different geographical regions in the world, it's important to mention that the LSM within ERA5/Land is forced with precipitation from the Numerical Weather Prediction (NWP) model used within ERA5. NWP models fundamentally struggle to capture precipitation in the tropics, and this will impact the results here. See for example Lavers et al. (2022).

We thank the referee for the reference and the point raised. We have added that to Section 4 on the third paragraph as: "Moreover, the ERA5-Land reanalysis is driven with precipitation from ERA5, known to have biases in the tropics Lavers et al. (2022), which could lead to biases in the runoff of ERA5-Land in these regions. As a consequence, our river model could be learning to correct these biases in addition to routing water. This is a possible pitfall for machine learning models trained with model output and not with observed data."

**Table A1:** Key details missing. Which datasets do each variable come from? What time period, temporal resolution is used?

We added the source for the data presented in the table in Appendix's A text. It's important to notice that all these variables are static attributes and are supposed not to have time dependence.

**Minor points**

Pg2, para 1, line 5: please change "regions is" to "regions are".
We corrected this.

Pg2, para2, line 5: suggest changing "forming the streamflow" to "forming streamflow".
We rephrased this.

Pg 3, bullet iv: Change "run operationally by the European Copernicus program" to something like, "GloFAS, the European Union Copernicus Emergency Management Service (CEMS) global flood forecasting system run operationally at the European Centre for Medium-Range Weather Forecasts (ECMWF)".
We corrected that after suggestions made by the GloFAS team in private conversations.

Pg8, line 2: missing the description of "Xs"
We have added an explicit description of x^d_t and x^s.

Pg 9, line 7-9 (I think, there are no line numbers included!): An NSE > 0 or KGE > ~-0.41 is not "good". The interpretation is that the model is performing better than a mean flow benchmark. This is what NSE=0 or KGE=~-0.41 means. It shows your model provides some level of skill beyond a very naïve mean flow benchmark. This should not be confused as "good". But great that you use the ~-0.41 threshold for the KGE, I agree with this!

We apologize for the line numbers. This happened because we posted our preprint both via EGU and via arXiv. EGU recommends having a single preprint, but arXiv doesn't allow papers with line numbers. We agree with the referee concerning the use of the word "good" and we have rephrased the text to avoid it.

Pg 14, Sect. 3.3, line 1: please change "present some simulated" to "present simulated".
We corrected this

Pg 18; last line: please change "doesn't" to "does not".
We corrected this.

Pag 19; line 2: You say routing in LSM component in climate models – but it's much wider than that. NWP models have a LSM, so this work is also relevant for short, medium and longer range hydrological forecasting using runoff from land surface models within weather models. It has a much wider impact that just the climate models!
We agree with the referee. While our main goal is to integrate this model to a climate model, it's true that it can also be used within the LSM of any weather forecasting model, as long as it has an LSM component that explicitly accounts for runoff. We have modified the text throughout to include this.